# Nonlinear photon-atom coupling with 4Pi microscopy

Yue-Sum Chin [1], Matthias Steiner [1,2] & Christian Kurtsiefer [1,2]

Implementing nonlinear interactions between single photons and single atoms is at the forefront of optical physics. Motivated by the prospects of deterministic all-optical quantum logic, many efforts are currently underway to find suitable experimental techniques. Focusing the incident photons onto the atom with a lens yielded promising results, but is limited by diffraction to moderate interaction strengths. However, techniques to exceed the diffraction limit are known from high-resolution imaging. Here we adapt a super-resolution imaging technique, 4Pi microscopy, to efficiently couple light to a single atom. We observe 36.6(3)% extinction of the incident field, and a modified photon statistics of the transmitted field–indicating nonlinear interaction at the single-photon level. Our results pave the way to few-photon nonlinear optics with individual atoms in free space.

---

[1] Centre for Quantum Technologies, 3 Science Drive 2, Singapore 117543, Singapore. [2] Department of Physics, National University of Singapore, 2 Science Drive 3, Singapore 117542, Singapore. Correspondence and requests for materials should be addressed to C.K. (email: christian.kurtsiefer@gmail.com)

To realize nonlinear interactions between a few propagating photons and a single atom in free space, the photons need to be tightly focused to a small volume[1–8]. From high-resolution imaging, it is well-known that a small focal volume requires optical elements, which cover a large fraction of the solid angle[9]. While standard confocal optical microscopy accomplished already very small probe volumes, the excitation light is focused through a lens that can cover only up to half of the solid angle, limiting the axial resolution due to a focal volume elongated along the optical axis. This limitation has been overcome by using two opposing lenses with coinciding focal points, known as 4Pi arrangement[10] The path of the incident beam is split, and the object is coherently illuminated by two counter-propagating parts of the field simultaneously (Fig. 1a). In this way, the input mode covers almost the entire solid angle, limited only by the numerical aperture of the focusing lenses.

The symmetry between imaging and excitation of quantum emitter suggests that a 4Pi arrangement can also be used to efficiently couple light to an atom. This intuitive argument is confirmed by numerical simulations of the electric field distribution near the focal point, from which we obtain the light-atom coupling efficiency $\Lambda = |E_{input}|^2 / |E_{max}|^2$, where $E_{input}$ is the electric field amplitude parallel to the atomic dipole and $E_{max}$ is the maximal amplitude of a pure dipole wave with the same power as the incident field (Fig. 1b–e)[11, 12]. The coupling efficiency $\Lambda$ can be understood as a geometric quantity describing the spatial mode overlap between the atomic dipole mode and the input mode, with $\Lambda = 1$ corresponding to complete spatial mode overlap.

In this work, we measure the photon-atom interaction strength for a single atom illuminated in the 4Pi arrangement. The strong increase in light scattering compared to one-sided illumination demonstrates a close to two-fold increase in interaction strength. The 4Pi arrangement leads to a sizeable nonlinearity of the interaction at the single-photon level which is manifested in the intensity correlations of the transmitted field.

## Results

**Experimental setup**. In our experiment, we hold a single $^{87}$Rb atom between two lenses with a far-off-resonant optical dipole trap (FORT) operating at a wavelength 851 nm[13]. We compare 4Pi and one-sided illumination by performing a transmission experiment with a weak coherent field driving the closed transition $5S_{1/2}$, $F = 2$, $m_F = -2$ to $5P_{3/2}$, $F = 3$, $m_F = -3$ near 780 nm[14]. The power of the probe field is well below the saturation power $P_{sat}$ of the corresponding transition, which is set to approximately $0.003 P_{sat}$. The probe beam originates from a collimated output of a single mode fiber. After splitting into path 1 and path 2, the beam is focused onto the atom through lenses $L_1$ and $L_2$ (Fig. 1a). The opposing lens re-collimates the probe beam, which is then coupled via an asymmetric beam splitter into a single mode fiber connected to avalanche photodetector $D_1$ or $D_2$, respectively (Supplementary Note 1). The total electric field of the light moving away from the atom is a superposition of the probe field and the field scattered by the atom[11, 15]. We denote the respective electric field amplitudes at the detectors, which is after the projection onto the mode of the optical fiber, $E_p$ for the probe, and $E_{sc}$ for the scattered field. In the limit of weak excitation, the atom reacts to the parts of the probe field propagating in path 1 and 2 independently. Consequently, the scattered field consists of two contributions $E_{sc} = E_{sc,1} + E_{sc,2}$. At detector $D_1$, the total electric field is the sum of the transmitted probe field in path 1, $E_p = \sqrt{P_{1,in}}$, and the scattered field contributions $E_{sc,1} = -2\Lambda_1 \sqrt{P_{1,in}}$ and $E_{sc,2} = -2\sqrt{\Lambda_1 \Lambda_2} \sqrt{P_{2,in}}$, where $P_{1(2),in}$ is the optical power, and $\Lambda_{1(2)}$ is the light-atom coupling efficiency of path 1(2). We further assume that the two counter-propagating probe fields have the same phase at the position of the atom. The power $P_1$ at detector $D_1$ is then given by

$$P_1 = \left( \sqrt{P_{1,in}} - 2\Lambda_1 \sqrt{P_{1,in}} - 2\sqrt{\Lambda_1 \Lambda_2} \sqrt{P_{2,in}} \right)^2. \quad (1)$$

Similarly, the power at detector $D_2$ is obtained by exchanging subscripts 1 and 2. From Eq. 1, we obtain the expected values for the individual transmission $T_{1(2)} = P_{1(2)}/P_{1(2),in}$, and the total transmission $T_{total} = (P_1 + P_2)/(P_{1,in} + P_{2,in})$. For example, for a one-sided illumination through lens $L_1$, that is, $P_{2,in} = 0$, the transmission measured at detector $D_1$ takes the well-known expression $T_1 = (1 - 2\Lambda_1)^2$[11, 15]. In the 4Pi configuration, we determine the total coupling $\Lambda_{total}$ from the total transmission $T_{total} = (1 - 2\Lambda_{total})^2$. From eq. 1, we find that the power splitting $P_{2,in} = P_{1,in}\Lambda_2/\Lambda_1$ optimizes the total coupling to $\Lambda_{total} = \Lambda_1 + \Lambda_2$.

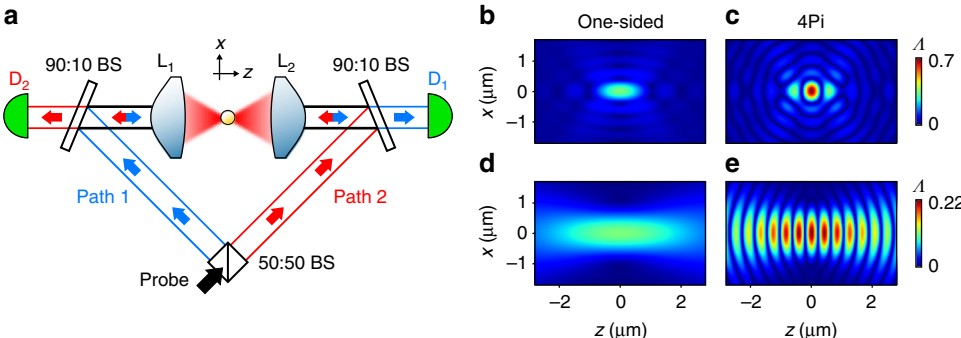

**Fig. 1** Concept of 4Pi illumination. **a** Schematics of the optical setup. The probe beam (black arrow) is split into path 1 (blue arrows) and path 2 (red arrows). The two beams then illuminate the atom from counter-propagating directions. Asymmetric beam splitters are used to sample the probe light after passing the atom. The probe light in path 1(2) is coupled into a single mode fiber connected to detector $D_{1(2)}$. By blocking one path, we recover the commonly employed one-sided illumination. BS: beam splitter, $L_{1(2)}$: high numerical aperture lens, $D_{1(2)}$: avalanche photodetector. **b–e** Numerical results of the coupling efficiency $\Lambda$ near the focal point. We consider a Gaussian field resonantly driving a circularly polarized dipole transition near 780 nm and evaluate the electric field distribution according to ref. [11], which includes the spatially varying polarization of the tightly focused probe light near the focus. The field is assumed to constructively interfere at the focal point for the 4Pi configuration. **b**, **c** Focusing parameters corresponding to an objective with numerical aperture NA = 0.93 and an input beam waist which experiences less than 1% clipping losses from the aperture of the lens. **d**, **e** Focusing parameters used in this experiment (NA = 0.75, input beam waist $w_0 = 2.7$ mm at lens, focal length $f = 5.95$ mm)

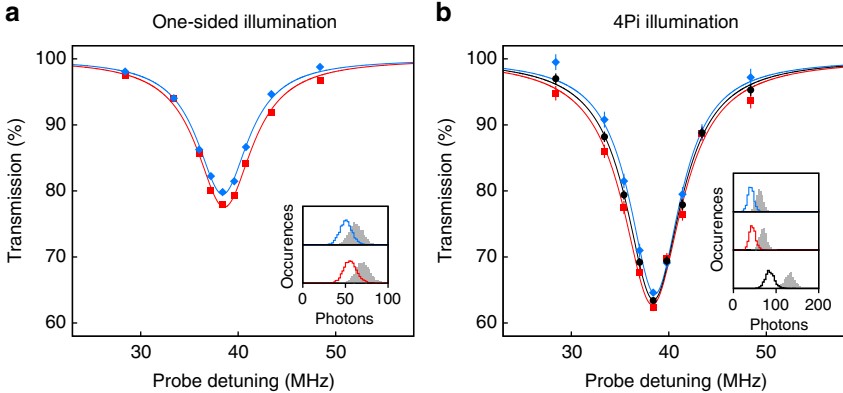

**Fig. 2** Extinction of a weak coherent probe beam. **a** One-sided illumination. Transmission at detector $D_1$ (blue diamonds) and $D_2$ (red squares) when probing via path 1 or path 2, respectively. Solid lines are Lorentzian fits. The inset shows the normalized histogram of detected photons during the probe cycle (solid line) and reference cycle (grey) for the resonant data point. **b** Same as **a** but with 4Pi illumination. The total transmission (black circles) is obtained from the sum of detectors $D_1$ and $D_2$. Error bars represent ±1 s.d. of propagated Poissonian counting uncertainties. The far-off-resonant optical dipole trap (FORT) shifts the resonance frequency by ~38.5 MHz compared to the natural transition frequency

**Transmission experiment**. Figure 2a shows the transmission spectrum of a weak coherent field for one-sided illumination, either via path 1 (blue) or path 2 (red). Comparing the resonant transmission $T_1 = 79.8(3)\%$ and $T_2 = 77.9(2)\%$ to eq. 1, we find similar coupling efficiencies, $\Lambda_1 = 0.053(1)$ and $\Lambda_2 = 0.059(1)$, as expected for our symmetric arrangement with two nominally identical lenses. Therefore, the maximum coupling expected in the 4Pi configuration is $\Lambda_{\text{total}} = \Lambda_1 + \Lambda_2 = 0.112(4)$, assuming perfect phase matching of the fields and ideal positioning of the atom.

In the 4Pi configuration, the atom needs to be precisely placed at an anti-node of the incident field (Fig. 1e). To this end, we tightly confine the atom along the optical axis with an additional blue-detuned standing wave dipole trap (761 nm). As the atom is loaded probabilistically into the optical lattice, we use a simple postselection technique to check whether the atom is trapped close to an anti-node of the incident field (see "Methods" section). Figure 2b shows the observed transmission when the atom is illuminated in the 4Pi arrangement. The increased light-atom coupling is evident from the strong reduction of transmission. The individual transmissions $T_1 = 64.6(5)\%$, $T_2 = 62.3(5)\%$, and the total transmission $T_{\text{total}} = 63.4(3)\%$ are significantly lower compared to the one-sided illumination. The corresponding total coupling of $\Lambda_{\text{total}} = 0.102(1)$ is close to the theoretical prediction of 0.112(4).

We next show that for a symmetric arrangement $\Lambda_1 \approx \Lambda_2$, the highest interaction is achieved with an equal power splitting $P_{2,\text{in}} \approx P_{1,\text{in}}$. Figure 3 displays the resonant transmissions for different relative beam power in the two paths. For imbalanced beam power, the total transmission is increased, albeit with a fairly weak dependence. In contrast, we find a strong dependence of the individual transmissions on the relative beam power: For $P_{1,\text{in}} \approx 12 P_{2,\text{in}}$, the total transmission is still low, $T_{\text{total}} = 71.2(8)\%$, but the two values for the individual transmissions are no longer equal: $T_{1,4\text{Pi}} = 74.0(8)\%$, $T_{2,4\text{Pi}} = 41(2)\%$. Figure 3 (solid lines) also shows that the observed behavior of the transmission is well reproduced by Eq. 1 without free parameter. However, the measured transmission values are mostly larger than expected from Eq. 1 due to the thermal motion of the atom[14] and the limited resolution of selecting the atom position.

**Photon statistics of transmitted light**. The nonlinear character of the photon-atom interaction can induce effective attractive or repulsive interactions between two photons[16–19]. These

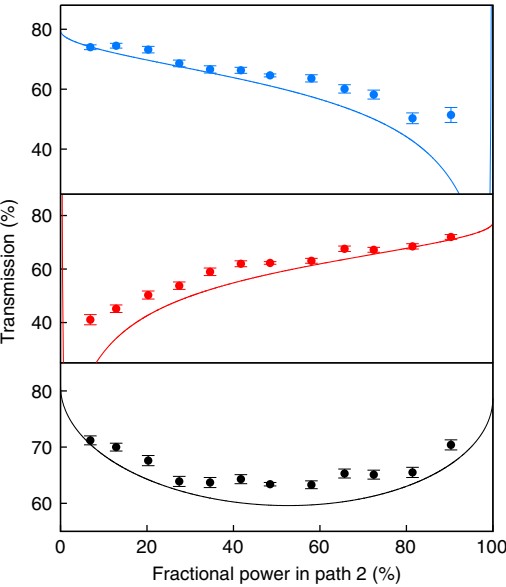

**Fig. 3** Resonant transmission for different power splittings between path 1 and path 2. Transmission at detector $D_1$ (top), $D_2$ (center) and the total transmission $D_1+D_2$ (bottom). The total number of incident photons is kept constant. Solid lines are $T_{1(2)}$ and $T_{\text{total}}$ derived from Eq. 1. Error bars represent ±1 s.d. of propagated Poissonian counting uncertainties

interactions can be observed as modification of the photon statistics of the transmitted field if the initial field contains multi-photon contributions[20–24]. A quantitative description of this effect has been developed in the context of waveguide quantum electrodynamics[25, 26]. For a weak coherent driving field, that is, ignoring contributions from number states with three or more photons, the second-order correlation function $g^{(2)}(\tau)$ takes the specific form

$$g^{(2)}(\tau) = e^{-\Gamma_0 \tau} \left( \left( \frac{2\Lambda}{1-2\Lambda} \right)^2 - e^{\frac{\Gamma_0 \tau}{2}} \right)^2, \qquad (2)$$

where $\Gamma_0 = 2\pi \times 6.07$ MHz is the excited state linewidth. By time-tagging the detection events at detector $D_1$ and $D_2$ during the probe phase, we obtain $g^{(2)}(\tau) = \langle p_1(t) p_2(t+\tau) \rangle / (\langle p_1(t) \rangle \langle p_2(t+\tau) \rangle)$, where $p_{1(2)}(t)$ is the detection probability at detector $D_{1(2)}$ at time $t$, and $\langle \rangle$ denotes the long time average. To acquire sufficient statistics,

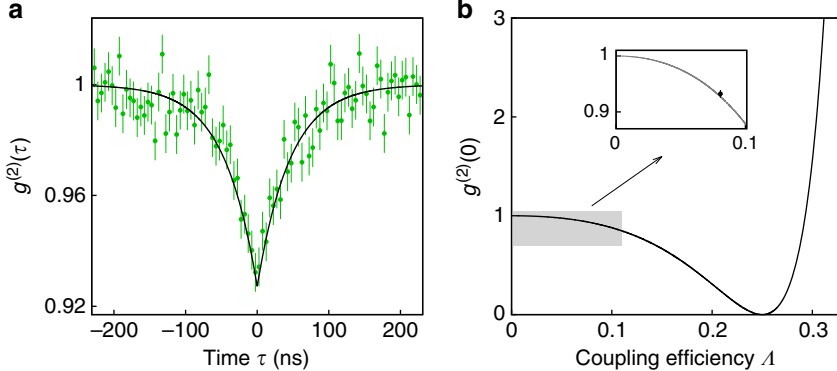

**Fig. 4** Modified photon statistics due to nonlinear interaction. **a** Intensity correlation of transmitted light with a time bin width of 5 ns. Solid line is the theoretical prediction without free parameter (Eq. 2). **b** Dependence on the coupling efficiency $\Lambda$. The inset is a zoom into the region of our data point for clarity, and the solid line is $g^{(2)}(0)$ from Eq. 2. Error bars represent $\pm 1$ standard error of mean

we use 50% more photons in the probe pulse as compared to Fig. 2, and also atoms which are not optimally coupled to the probe field (see "Methods" section). From the resulting average transmission $T_{\mathrm{total}} = 70.3(3)\%$, we deduce an average coupling $\Lambda_{\mathrm{total}} = 0.0808(5)$ for this experiment. As shown in Fig. 4, we find a clear signature of nonlinear photon-atom interaction in the intensity correlations of the transmitted light. The observed photon anti-bunching $g^{(2)}(0) = 0.934(7)$ is in good agreement with Eq. 2. Here, for fair comparison with Eq. 2, we account for a small photon bunching effect ($\approx 1.7\%$, see "Methods" section) due to the diffusive atomic motion[27, 28]. For stronger light-atom coupling the changes of the photon statistics are expected to be more significant (Fig. 4b). Notably, for $\Lambda = 0.25$ the transmitted and the reflected light shows perfect anti-bunching ($g^{(2)}(0) = 0$), that means the atom acts as a photon turnstile converting a coherent field completely into a single-photon field. The transmission for this light-atom coupling is $T_{\mathrm{total}} = 25\%$ (Eq. 1). Photon bunching ($g^{(2)}(0) > 1$) for large values of $\Lambda$ signals an enhanced transmission probability when two photons are simultaneously incident; while one photon states are efficiently reflected, photon pairs saturate the atomic transition and have a larger transmission probability.

## Discussion

Our work establishes the 4Pi arrangement as an effective technique to couple a propagating field to an atom. This opens exciting prospects to implement effective interactions between photons with tightly focused free space modes and single atoms. Strongly interacting photons could find application in imaging, metrology, quantum computing and cryptography, and constitute a novel platform to study many-body physics[29, 30]. The presented approach forms an experimental alternative to waveguide/cavity quantum electrodynamics[20, 31] and Rydberg quantum optics[24, 32–34]. While the achieved nonlinearity of the photon-atom interaction, observed as modification of the photon statistics, does not create strongly correlated photons, yet, the 4Pi arrangement eases the technical requirements to the focusing lens considerably, making the implementation of strong photon-photon interaction feasible. In the near future, we expect that by using higher numerical aperture lenses, the 4Pi arrangement will enable $\Lambda = 0.25$, and thus the efficient conversion of a coherent beam into single photons (Eq. 2). Even stronger interactions ($\Lambda \approx 0.7$) are technically within reach with state-of-the-art objectives in 4Pi arrangement[35]. Finally, we note that to use the 4Pi configuration for quantum technological applications, the photon loss due to the asymmetric beam splitters can be avoided

by probing directly the output port of the 50:50 beam splitter, shown in Fig. 1a.

## Methods

**Measurement sequence and postselection of the atom position**. The experimental sequence starts with loading a single atom from a cold ensemble in a magneto-optical trap into a far-off-resonant dipole trap. Once trapped, the atom undergoes molasses cooling for 5 ms[36]. We then apply a bias magnetic field of 0.74 mT along the optical axis and optically pump the atom into the $5S_{1/2}$, $F = 2$, $m_F = -2$ state. Subsequently, we perform two transmission experiments during which we switch on the probe field for 1 ms each. The power of the probe field is ~$0.003 P_{\mathrm{sat}}$, that is, well below the saturation power $P_{\mathrm{sat}}$ of the corresponding transition. We tune the frequency of the first probe pulse to obtain the transmission spectra shown in Fig. 2. The second probe pulse is used to check whether the atom has been trapped at an anti-node of the probe field. The frequency of the probe field during the second probe pulse is set to be resonant with the atomic transition. To obtain the relative transmission, we also detect the instantaneous probe power for each transmission experiment by optically pumping the atom into the $5S_{1/2}$, $F = 1$ hyperfine state, which shifts the atom out of resonance with the probe field by 6.8 GHz, and reapplying the probe field.

The position of the atom is postselected based on the detected transmission during the second probe cycle. For an atom loaded into a desired site of the potential well, the transmission is low. Hence, we discard detection events in the first probe cycle if the number of photons detected in the second cycle is above a threshold value. For the data shown in Figs. 2b and 3, we use a threshold that selects ~0.5% of the total events as a trade-off between data acquisition rate and selectivity of the atomic position. To measure the second-order correlation function of the transmitted light (Fig. 4a), we choose a higher threshold which selects about 10% of the experimental cycles. For the case of one-sided illumination, this postselection procedure does not change the observed transmission (Supplementary Note 2).

**Normalization of second-order correlation function**. We measure the second-order correlation function of the transmitted light using detector $D_1$ and $D_2$ as the two detectors of a Hanbury–Brown and Twiss setup. The photodetection events are time tagged during the probe phase, and sorted into a time delay histogram. We obtain the normalized correlation function $g^{(2)}(\tau)$ by dividing the number of occurrences by $r_1 r_2 \Delta t T$, where $r_{1(2)}$ is the mean count rate at detector $D_{1(2)}$, $\Delta t$ is the time bin width, and $T$ is the total measurement time. For times 100 ns $< \tau < 1$ μs, we find super-Poissonian intensity correlations $g^{(2)}(\tau) > 1$, which are induced by the atomic motion through the trap. Although the amplitude of the correlations is small, we nevertheless perform a deconvolution for a better comparison to Eq. 2. The correlations are expected to decay exponentially for diffusive motion, thus we fit $f(\tau) = 1 + a_0 \exp(-\tau/\tau_d)$ to $g^{(2)}(\tau)$, resulting in decay time constant $\tau_d = 0.71(8)$ μs and amplitude $a_0 = 0.019(2)$. Figure 4 shows the second-order correlation function after deconvolution of the diffusive motion that is after division by $f(\tau)$ (see Supplementary Note 3).

**Data availability**. The data that support the findings of this study are available from the corresponding author upon reasonable request.

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

## Acknowledgements

We acknowledge support of this work by the Ministry of Education in Singapore (AcRF Tier 1) and the National Research Foundation, Prime Minister's office. M. Steiner acknowledges support by the Lee Kuan Yew Postdoctoral Fellowship.

## Author contributions

Y.-S.C. and M.S.: Performed the experiments and data analysis. M.S. and C.K.: Conceived the experiment. All authors discussed the result and participated in writing the paper.

## Additional information

**Competing interests:** The authors declare no competing financial interests.

