## [Peer Review File · Nature Communications]

Reviewers' comments:

Reviewer #1 (Remarks to the Author):

In this manuscript, the authors report the coupling of a few-photon light field to a single atom in free-space making use of a "4pi" lens configuration, as known from high-resolution imaging. The achieved extinction of probe photon transmission due to scattering from the single atom reaches 37%, a value comparable to state-of-the-art waveguide systems.

The presented results are an interesting step forward in the domain of freespace-QED. Both the experiment as well as the analysis and comparison to theory are nicely presented and interesting, I only have a few comments which are listed below. I think this work deserves publication in Nat. Comm. after the authors have considered these comments.

* I think the very limited length of the manuscript stems from previous restrictions. As I understand it in NatComm these restrictions are much less severe. I would highly appreciate some more detail at various places in the manuscript. Most importantly to me are the presentations of eq. 1 and 2, which now are simply given together with references. A short overview of the derivation and/or somewhat more detailed discussion would be very useful to a reader not extremely familiar with the topic.

* I find the repeatedly used phrase "weak probe beam" somewhat unclear. What exactly is meant by weak here? That the effective coupling strength (per time) is small compared to the excited state decay rate Γ ?

This more quantitative definition should be particularly important for the validity of eq. (2). In case of the coupling strength exceeding the decay rate, the single TLS should undergo Rabi oscillations, as recently observed for a Rydberg superatom in <https://arxiv.org/abs/1705.04128>. The theory presented there gives the full expression for $g_2(t_1, t_2)$ for any coupling strength. Eq. (2) should be valid in the "weak coupling regime".

* The agreement between theory and experiment in Fig. 3 is not too convincing (it is called "qualitative" in the text). The authors seem to understand their system very well, as e.g. shown by the nice analysis of the g_2 data. Why does this (more basic) measurement of the transmission not agree so well? A bit of discussion would be good here.

* I do not fully understand the discussion of the "single-photon converter" at the end of the manuscript, reachable with a stronger coupling. Do the authors mean that for stronger coupling a single photon is reflected while all others are transmitted? Such a "single-photon subtractor" has recently been demonstrated with a single atom coupled to a nanofiber (Nature Photonics 10, 19-22), although the physics of this system is slightly different as it makes use of 3 atomic levels and the polarization dependence of the light in the fiber. A somewhat more detailed discussion how this scheme would work for the single TLS would be nice.

* A question in connection to these nanofiber systems: with the tight focussing of the light, to what extent is the field still transverse? Or does spatially varying polarization already play a role here, as in the (sub-wavelength) fiber-systems? Will that be more relevant for the future higher NA system mentioned in the outlook?

Reviewer #2 (Remarks to the Author):

In their manuscript "Nonlinear photon-atom coupling with 4Pi microscopy" the authors implement a technique known from high-resolution imaging to increase the atom-light coupling. Principle of this

technique is to illuminate the atom from opposite directions such that the driving field forms a standing wave. Then, the atom-photon coupling is strongly increased if the atom is held in an antinode of the driving field. The authors observe an extinction of the driving field by about 36 %, a value that has not been reached before. Furthermore, they demonstrate in a photon-correlation measurement that the nonlinear interaction between atom and photon leads to a modified photon statistics in the transmitted field.

The experimental claims are convincing, the manuscript well written and the representation of the experimental results clear. The results are important for other groups and will certainly motivate them to increase their emitter-photon coupling by using this technique. I recommend publication in Nature Communications.

However, there are a few (minor) points the authors should address before publication:

- For easier comparison of Fig. 1 b/c with d/e, it would be nice if the authors give the NA of their lenses in the figure caption. It is only given in the Supplementary Information.
- If space constrains allow it, it would be helpful for the general reader to explain the terms of Eq. 1 intuitively.
- There must be mistake in the optimal power splitting, it must read $P_{2,in} = P_{1,in} \frac{\lambda_2}{\lambda_1}$
- Can the authors comment on how the total coupling λ_{total} depends on the chosen post selection? Could they increase the measured value by an even stricter post selection, or is the residual difference due to atomic motion in the 1D lattice? Could the value reach the theoretical limit if a 3D lattice would be used?
- It does not become clear to me why larger values of λ will lead to photon bunching in the transmitted field. Could the authors explain this a bit more?
- Is a value of $\lambda=0.25$ within experimental reach? What NA lenses would be necessary?
- The authors motivate their study with the prospect of deterministic all optical quantum logic. Here, I see a weak point in the 4π technique: one has to use beam splitters to separate the input mode from the output mode. Therefore, there is always a trade of between the fraction of input light that is sent towards the atom and the fraction of light that can be detected. In an all-optical quantum processor, it might be important to not lose input light as well as output light. Have the authors thought about this problem? Have they ideas how to deal with it? It would be great if they could include a short discussion on this in the outlook.
- Methods: The post selection process does not become clear only from this paragraph, I could only understand it after reading the SI. It is not mentioned here, that the transmission in the second interval must be below a certain value, which signals good photon-atom coupling, in order to take data in the first interval into account.
- SI: on page 3, instead of referencing to Fig3, they have to reference to Fig. 4. This mistake has been made twice.
- SI: normalization of $g^{(2)}(\tau)$. It seems to me that the normalization of each interval makes the data quite noisy, since there are only about 150 counts per interval with an error of +/-12 due to

counting statistics. I expect that the fluctuations due to movement of the optical lattice are slow, so maybe it would make more sense to normalize about 20 intervals together in order to reduce noise? Have the authors considered this?

Reviewer #3 (Remarks to the Author):

The manuscript "Nonlinear photon-atom coupling with 4pi microscopy" reports on experiments where light is transmitted through/by a single atom held at the tight focus of the laser beam. The main findings are that double-sided illumination decreases the transmission significantly, and that the transmitted light shows sub-Poissonian photon statistics. I consider these important experimental demonstrations in the field, and well supported by the data presented, indicating that the conclusions are valid. The findings are clearly beyond statistical fluctuations and further supported by comparison to simple theoretical predictions. In the case of the sub-Poissonian photon statistics the "unprocessed data" is also shown in the supplementary information, which removes any doubt that conclusions are drawn based on artifacts from the data-processing. The topic is of high interest to the scientific community and will definitely be of interest outside its specific field. The manuscript is well written in clear English. I enjoyed reading it. I am therefore happy to recommend publication in Nature Communications.

I have few minor comments:

1. Where the light-atom coupling efficiency Λ is introduced, I did not understand its definition without looking in Ref 15. In particular what the maximal possible amplitude referred to. As Λ plays a central role in the paper it would be good to clarify this.
2. As far as I can see, it is only said in the supplementary material, that for one-sided illumination, the postselection procedure does not change the observed transmission. This is a crucial point for interpretation of the main data, so I suggest that this is stated in the main text or methods.
3. The caption of Fig. 2 is a bit misleading. I presume that red and blue in b refers to the two detectors rather than illumination paths as is indicated.

Point-to-Point response to referee comments:

Reviewer A comment :

1) I think the very limited length of the manuscript stems from previous restrictions. As I understand it in NatComm these restrictions are much less severe. I would highly appreciate some more detail at various places in the manuscript. Most importantly to me are the presentations of eq. 1 and 2, which now are simply given together with references. A short overview of the derivation and/or somewhat more detailed discussion would be very useful to a reader not extremely familiar with the topic.

Reply: We expanded the discussion of Eq. 1 and explained the individual terms. Regarding Eq. 2, we included a statement about the weak power assumption.

Section: Results – Experimental Setup

"The total electric field of the light moving away from the atom is a superposition of the probe field and the field scattered by the atom [11,15]. We denote the respective electric field amplitudes at the detectors, that is after the projection onto the mode of the optical fibre, E_p for the probe, and E_{sc} for the scattered field. In the limit of weak excitation, the atom reacts to the parts of the probe field propagating in path 1 and 2 independently. Consequently, the scattered field consists of two contributions $E_{sc}=E_{sc,1}+E_{sc,2}$. At detector D_1, the total electric field is the sum of the transmitted probe field in path 1, $E_p=\sqrt{P_{1,in}}$, and the

scattered field contributions $E_{\text{sc},1} = -2 \Lambda_1 \sqrt{P_{1,\text{in}}}$ and $E_{\text{sc},2} = -2 \sqrt{\Lambda_1 \Lambda_2} \sqrt{P_{2,\text{in}}}$, where $P_{1(2),\text{in}}$ is the optical power, and $\Lambda_{1(2)}$ is the light-atom coupling efficiency of path 1(2). We further assume that the two counter-propagating probe fields have the same phase at the position of the atom. The power P_1 at detector D_1 is then given by...Eq. 1"

Section: Results - Photon statistics of transmitted light

"A quantitative description of this effect has been developed in the context of waveguide quantum electrodynamics [25, 26]. For a weak coherent driving field, i.e., ignoring contributions from number states with three or more photons, the second-order correlation function $g^{(2)}(\tau)$ takes the specific form ... Eq.2"

Reviewer A comment:

2) I find the repeatedly used phrase "weak probe beam" somewhat unclear. What exactly is meant by weak here? That the effective coupling strength (per time) is small compared to the excited state decay rate Γ ? This more quantitative definition should be particularly important for the validity of eq. (2). In case of the coupling strength exceeding the decay rate, the single TLS should undergo Rabi oscillations, as recently observed for a Rydberg superatom in <https://arxiv.org/abs/1705.04128>. The theory presented there gives the full expression for $g_2(t_1, t_2)$ for any coupling strength. Eq. (2) should be valid in the "weak coupling regime".

Reply: The probe power is well below the saturation power and corrections to Eq. (2) due to terms with more than 2 photons are negligible. We added an statement in Methods about the power of the probe beam. We also added the reference.

Section: Methods - Measurement sequence and postselection of the atom position
"The power of the probe field is approximately $0.003P_{\text{sat}}$, i.e., well below the saturation power P_{sat} of the corresponding transition."

Reviewer A comment:

3) The agreement between theory and experiment in Fig. 3 is not too convincing (it is called "qualitative" in the text). The authors seem to understand their system very well, as e.g. shown by the nice analysis of the g_2 data. Why does this (more basic) measurement of the transmission not agree so well? A bit of discussion would be good here.

Reply: In the experiment shown in Fig. 3 we vary the ratio of the beam power in the two illumination paths by almost two orders of magnitude. We compare our results to Eq.1 without any free parameter, instead Eq.1 only depends on the interaction strength measured with one-sided illumination. With this in mind, we find that the agreement of experiment and theory is actually quite good. We believe that the residual discrepancy, i.e., the fact that the observed interaction strength in the 4π arrangement is not the perfect sum of the two coupling efficiencies, originates from two factors: 1) Finite temperature of the atom which is more relevant in the 4π arrangement than for one-sided illumination and 2) the limited resolution with which we can select the atom position with respect to the probe field. We added a corresponding statement to the manuscript:

Section: Results - Transmission experiment

"Figure 3 (solid lines) also shows that the observed behaviour of the

transmission is well re-produced by equation 1 without free parameter. However, the measured transmission values are mostly larger than expected from equation 1 due to the thermal motion of the atom [14] and the limited resolution of selecting the atom position."

Reviewer A comment:

4) I do not fully understand the discussion of the "single-photon converter" at the end of the manuscript, reachable with a stronger coupling. Do the authors mean that for stronger coupling a single photon is reflected while all others are transmitted? Such a "single-photon subtractor" has recently been demonstrated with a single atom coupled to a nanofiber (Nature Photonics 10, 19-22), although the physics of this system is slightly different as it makes use of 3 atomic levels and the polarization dependence of the light in the fiber. A somewhat more detailed discussion how this scheme would work for the single TLS would be nice.

Reply: Our "single-photon converter" is different from the mentioned single-photon subtractor" as it converts a continuous weak coherent field into a single photon field. Importantly, the derivation of Eq.2 assumes a 'weak coherent field' which means that contributions from number states with 3 or more photons are neglected. Then for $\Lambda=0.25$, the number states with 1 photon have a finite probability to be transmitted or reflected while 2 photon terms are separated into one transmitted and one reflected photon. In contrast, the "single-photon subtractor" operates in a pulsed-mode and takes exactly one photon out of a pulse containing many photons. To clarify this, we included a statement about the assumptions for Eq.2 and state in the discussion that we are referring to a particular coupling strength, $\Lambda=0.25$.

Section: Results - Photon statistics of transmitted light

"For a weak coherent driving field, i.e., ignoring contributions from number states with three or more photons, the second-order correlation function $g^{(2)}(\tau)$ takes the specific form ..."

Section: Results - Discussion

"In the near future, we expect that by using higher numerical aperture lenses, the 4Pi arrangement will enable $\Lambda = 0.25$ and thus the efficient conversion of a coherent beam into single photons (see equation 2)."

Reviewer A comment:

5) A question in connection to these nanofiber systems: with the tight focussing of the light, to what extent is the field still transverse? Or does spatially varying polarization already play a role here, as in the (sub-wavelength) fiber-systems? Will that be more relevant for the future higher NA system mentioned in the outlook?

Reply: The local polarization of the electric field in the focus and in particular at the position of the atom is very important for the light-matter interaction. These effects are included in the numerical results of the coupling efficiency shown in Figure 1. We use a full vectorial model described in Tey et al. (now Ref.11), which derives the electric field polarization in the focus for a circularly polarized Gaussian beam. We make it explicit in the revised manuscript that the polarization effects are included with the following sentence in the caption of Figure 1:

"Numerical results of the coupling efficiency Λ near the focal point. We

consider a Gaussian field resonantly driving a circularly polarized dipole transition near 780 nm and evaluate the electric field distribution according to ref. [11], which includes the spatially varying polarization of the tightly focused probe light near the focus."

Reviewer B comment:

1) For easier comparison of Fig. 1 b/c with d/e, it would be nice if the authors give the NA of their lenses in the figure caption. It is only given in the Supplementary Information.

Reply: We included the NA of our lenses in the caption of Figure 1.

Reviewer B comment:

2) If space constrains allow it, it would be helpful for the general reader to explain the terms of Eq. 1 intuitively.

Reply: We expanded the discussion of Eq.1, explaining the individual terms. See details of the replies for Reviewer A comment 1) for the section Results - Experimental Setup ("The total electric field.....Eq. 1).

Reviewer B comment:

3) There must be mistake in the optimal power splitting, it must read $P_{\{2,in\}} = P_{\{1,in\}} \frac{\lambda_2}{\lambda_1}$

Reply: We corrected the mistake.

Reviewer B comment:

4) Can the authors comment on how the total coupling $\lambda_{\{total\}}$ depends on the chosen post selection? Could they increase the measured value by an even stricter post selection, or is the residual difference due to atomic motion in the 1D lattice? Could the value reach the theoretical limit if a 3D lattice would be used?

Reply: Yes, we believe that the residual motion is a limiting factor. For stricter post selection we observe only minor improvement in the interaction strength. A 3D lattice is unlikely to improve the interaction strength as the radial confinement is already quite tight. We included a statement in the main text about the slight discrepancy between experiment and theory which we believe is due to the postselection and atom temperature. Furthermore, we added a figure in the supplementary information which shows the dependency of the transmission on threshold value for 4π and one-sided illumination.

statement in main text, Section: Results - Transmission experiment:
"Figure 3 (solid lines) also shows that the observed behaviour of the transmission is well reproduced by equation 1 without free parameter. However, the measured transmission values are mostly larger than expected from equation 1 due to the thermal motion of the atom [14] and the limited resolution of selecting the atom position."

Reviewer B comment:

5) It does not become clear to me why larger values of λ will lead to photon bunching in the transmitted field. Could the authors explain

this a bit more?

Reply: For larger values of Λ the transmission reduces. However, the transmission probability for one-photon states drops faster than for two-photon states, which results in photon-bunching in the transmitted light. We expanded our discussion of the behaviour in the manuscript:

Section: Results - Photon statistics of transmitted light

"Photon bunching ($g^{(2)}(0) > 1$) for large values of Λ signals an enhanced transmission probability when two photons are simultaneously incident; while one photon states are efficiently reflected, photon pairs saturate the atomic transition and have a larger transmission probability."

Reviewer B comment:

6) Is a value of $\Lambda=0.25$ within experimental reach? What NA lenses would be necessary?

Reply: Even values above $\Lambda=0.25$ are in principle reachable with current technology. We added a brief discussion:

"Even stronger interactions ($\Lambda \approx 0.7$) are technically within reach with state-of-the-art objectives in 4π arrangement [35]."

Reviewer B comment:

7) The authors motivate their study with the prospect of deterministic all optical quantum logic. Here, I see a weak point in the 4π technique: one has to use beam splitters to separate the input mode from the output mode. Therefore, there is always a trade off between the fraction of input light that is sent towards the atom and the fraction of light that can be detected. In an all-optical quantum processor, it might be important to not lose input light as well as output light. Have the authors thought about this problem? Have they ideas how to deal with it? It would be great if they could include a short discussion on this in the outlook.

Reply: One possible solution to this problem is to detect the light exiting the used port of the 50:50 beam splitter. Here, the atom acts as a highly nonlinear medium in a Sagnac-interferometer. We included this discussion in the outlook.

Section: Discussion

"Finally, we note that to use the 4π configuration for quantum technological applications, the photon loss due to the asymmetric beam splitters can be avoided by probing directly the output port of the 50:50 beam splitter, shown in Fig. 1a."

Reviewer B comment:

8) Methods: The post selection process does not become clear only from this paragraph, I could only understand it after reading the SI. It is not mentioned here, that the transmission in the second interval must be below a certain value, which signals good photon-atom coupling, in order to take data in the first interval into account.

Reply: We refined the corresponding paragraph and included more details about the postselection procedure.

Section: Methods - Measurement sequence and postselection of the atom position
"The position of the atom is postselected based on the detected transmission during the second probe cycle. For an atom loaded into a desired site of the potential well, the transmission is low. Hence, we discard detection events in the first probe cycle if the number of photons detected in the second cycle is above a threshold value. For the data shown in Fig. 2b and Fig. 3 we use a threshold which selects approximately 0.5% of the total events as a trade-off between data acquisition rate and selectiveness of the atomic position. To measure the second-order correlation function of the transmitted light (Fig. 4a), we choose a higher threshold which selects ~10% of the experimental cycles. For the case of one-sided illumination, this postselection procedure does not change the observed transmission."

Reviewer B comment:

9) SI: on page 3, instead of referencing to Fig3, they have to reference to Fig. 4. This mistake has been made twice.

Reply: The references are corrected.

Reviewer B comment:

10) SI: normalization of $g^{(2)}(\tau)$. It seems to me that the normalization of each interval makes the data quite noisy, since there are only about 150 counts per interval with an error of +/-12 due to counting statistics. I expect that the fluctuations due to movement of the optical lattice are slow, so maybe it would make more sense to normalize about 20 intervals together in order to reduce noise? Have the authors considered this?

Reply: We chose to normalize the g_2 of each experimental cycle to avoid averaging over slightly different levels of probe power and light-atom couplings. Averaging over these slow fluctuations results in a bias of the g_2 (extra bunching). As our g_2 -anti-bunching dip is quite small, any bias needs to be avoided. We expanded the discussion of our choice of normalization in the supplementary information.

Supplementary note 3:

"To make the normalization robust against intensity drifts of the probe power and cycle-to-cycle variations of the light-atom coupling, we perform the normalization for every 1 ms-long measurement cycle..."

Reviewer C comment:

1) Where the light-atom coupling efficiency Λ is introduced, I did not understand its definition without looking in Ref 15. In particular what the maximal possible amplitude referred to. As Λ plays a central role in the paper it would be good to clarify this.

Section: Introduction

Reply: We expanded and clarified the description of the coupling efficiency Λ in the introduction:

"This intuitive argument is confirmed by numerical simulations of the electric field distribution near the focal point, from which we obtain the light-atom coupling efficiency $\Lambda = |E_{\text{input}}|^2 / |E_{\text{max}}|^2$, where E_{input} is the electric field amplitude parallel to the atomic dipole and E_{max} is the maximal amplitude of a pure dipole wave with the same power as the incident field (Fig. 1b-f) [11, 12]. The coupling efficiency Λ can be understood as a

geometric quantity describing the spatial mode overlap between the atomic dipole mode and the input mode, with $\Lambda = 1$ corresponding to complete spatial mode overlap."

Reviewer C comment:

2) As far as I can see, it is only said in the supplementary material, that for one-sided illumination, the postselection procedure does not change the observed transmission. This is a crucial point for interpretation of the main data, so I suggest that this is stated in the main text or methods.

Reply: We included the corresponding statement into methods.

Section: Methods - Measurement sequence and postselection of the atom position
"For the case of one-sided illumination, this postselection procedure does not change the observed transmission."

Reviewer C comment:

3) The caption of Fig. 2 is a bit misleading. I presume that red and blue in b refers to the two detectors rather than illumination paths as is indicated.

Reply: Correct, we edited the caption to clarify that the colour-coding refers to the detectors rather than the paths.

"Transmission at detector D1 (blue diamonds) and D2 (red squares) when probing via path 1 or path 2, respectively."

REVIEWERS' COMMENTS:

Reviewer #1 (Remarks to the Author):

I think the authors did a very thorough job answering the questions and comments from all three referees.

My only (tiny) remaining comment is that I would put the clear statement of the probe power $P=0.003$ P_{sat} in the main text instead of the SI. This may be perfectly clear to the authors when they write "weak probe", but for a more general audience this clarification would be nice.

But independent of what the authors choose on this, I recommend publication of this paper in Nat. Comm.

Reviewer #2 (Remarks to the Author):

My questions and comments from the first review have been fully addressed in the new version. Now I am happy to recommend the manuscript for publication in Nature Communications.

Reviewer #3 (Remarks to the Author):

The authors have clarified the points raised in my first review, and I recommend publication.

Point-to-Point response to referee comments:

Reviewer #1 comment:

1) My only (tiny) remaining comment is that I would put the clear statement of the probe power $P=0.003 P_{\text{sat}}$ in the main text instead of the SI. This may be perfectly clear to the authors when they write "weak probe", but for a more general audience this clarification would be nice.

Reply:

We move the probe power statement to the main text, since the first mention of "weak probe" in the Experimental setup part of Results section.

"The power of the probe field is well below the saturation power P_{sat} of the corresponding transition, which is set to approximately $0.003P_{\text{sat}}$."